https://doi.org/10.1038/s41467-019-11885-4　　**OPEN**

# An all-photonic full color RGB system based on molecular photoswitches

Gaowa Naren[1], Chien-Wei Hsu[1], Shiming Li[1], Masakazu Morimoto[2], Sicheng Tang [3], Jordi Hernando[4], Gonzalo Guirado [4], Masahiro Irie[2], Francisco M. Raymo[3], Henrik Sundén [1] & Joakim Andréasson [1]

On-command changes in the emission color of functional materials is a sought-after property in many contexts. Of particular interest are systems using light as the external trigger to induce the color changes. Here we report on a tri-component cocktail consisting of a fluorescent donor molecule and two photochromic acceptor molecules encapsulated in polymer micelles and we show that the color of the emitted fluorescence can be continuously changed from blue-to-green and from blue-to-red upon selective light-induced isomerization of the photochromic acceptors to the fluorescent forms. Interestingly, isomerization of both acceptors to different degrees allows for the generation of all emission colors within the red-green-blue (RGB) color system. The function relies on orthogonally controlled FRET reactions between the blue emitting donor and the green and red emitting acceptors, respectively.

[1] Chemistry and Chemical Engineering, Chemistry and Biochemistry, Chalmers University of Technology, 41296 Göteborg, Sweden. [2] Department of Chemistry and Research Center for Smart Molecules, Rikkyo University Nishi-Ikebukuro 3-34-1, Toshima-ku, Tokyo 171-8501, Japan. [3] Laboratory for Molecular Photonics, Department of Chemistry, University of Miami, 1301 Memorial Drive, Coral Gables, FL 33146-0431, USA. [4] Departament de Química, Universitat Autònoma de Barcelona, 08193 Cerdanyola del Vallès, Spain. Correspondence and requests for materials should be addressed to J.A. (email: a-son@chalmers.se)

Stimuli-responsive multicolor luminescent materials attract extensive attention due to their potential applications in areas such as optical memory systems, bioimaging, sensors, white-light generation, encryption/decryption protocols, and anti-counterfeiting[1–9]. Additive mixing, that is, changes in the overall emission color of a sample achieved by mixing two or more individual fluorophores in different ratios has been successfully applied[10–14], whereas other approaches imply exposing the sample to physical or mechanical stimuli such as acid/base, ions, solvent vapor, and grinding/smearing[15–26]. The major downside with all the above mentioned efforts is that they require physical access to the sample. Thus, closed systems are excluded, as is remote operation. Using light instead as the external stimulus eliminates the need for physical access. It is also waste-free, non-invasive, and can be delivered with extremely high spatiotemporal precision (where and when). Several, at least partly, photonically driven systems have been devised to display a large range of attainable changes in the emission color[27–49], often by the inclusion of photochromic molecules (molecular photoswitches). Most appealing would be an all-photonic system that allows for full-color reproduction, that is, all colors in the visible spectrum can be generated from one and the same sample. This communication describes such a system. The function relies on orthogonally controlled FRET (Förster Resonance Energy Transfer) processes between a blue emitter (perylene) and two diarylethene (DAE) photoswitches emitting green and red light, respectively. Selective isomerization of the two photoswitches results in blue-to-green and blue-to-red emission color changes in a color-correlated[3] fashion, whereas isomerization of both photoswitches allows for the generation of all emission colors within the red-green-blue (RGB) color system.

## Results

**System design**. Figure 1 shows the structures and the isomerization scheme for perylene (per) and the two photochromic DAE derivatives[50].

The absorption spectra of all species are shown in Supplementary Fig. 2. We refer herein to the two DAE derivatives as DAEg and DAEr, the g and the r referring to the emission color of the closed isomers (green and red, respectively). Indicated are also the FRET processes responsible for the color changes. The open forms of the DAE photoswitches, DAEg(o) and DAEr(o),

display absorption only at wavelengths shorter than 450 nm and show no or only minute overlap with the blue per emission (Supplementary Fig. 2). UV light exposure isomerizes the DAEs to the closed isomers, displaying absorption also in the visible region of the spectrum. The absorption of both DAEg(c) and DAEr(c) overlaps substantially with the blue emission from per, implying that they can act as quenchers of the per emission in FRET processes. The FRET processes would not only imply quenching of the blue emission from per, but also sensitization of the fluorescence from the respective DAE(c) isomer: Green emission from DAEg(c) and red emission from DAEr(c). Thus, three extreme situations can be identified: (i) No UV-induced isomerization implies both DAE derivatives in the open isomeric form (DAEg(o) and DAEr(o)). Per is the sole fluorophore, and blue fluorescence is emitted. (ii) UV-induced isomerization of only DAEg(o) to yield DAEg(c) and DAEr(o). FRET from per to DAEg(c) results in green emission from DAEg(c). (iii) Isomerization of only DAEr(o) to yield DAEg(o) and DAEr(c). FRET from per to DAEr(c) results in red emission from DAEr(c). In all other intermediate situations (partial isomerization of one or both DAE:s and partial FRET reactions), the overall emission will be a blend between red (R), green (G), and blue (B), implying that full color reproduction in the RGB system would be possible.

Key to the overall function described above is selective, or orthogonal, isomerization of the open to the closed forms, DAEg(o) → DAEg(c) and DAEr(o) → DAEr(c). As indicated in Fig. 1, we have selected 242 and 415 nm for the respective reactions. From Supplementary Fig. 2, it is seen that DAEg(o) shows no detectable absorption at 415 nm, so that this wavelength is expected to trigger exclusively the DAEr(o) → DAEr(c) closing reaction. Although DAEg(o) is the stronger absorber at 242 nm, light of this wavelength is also absorbed by DAEr(o). However, as the isomerization quantum yield for the DAEg(o) → DAEg(c) reaction is substantially higher than for DAEr(o) → DAEr(c) (0.50 vs. 0.0005 in acetonitrile), 242 nm exposure is expected to be virtually specific for DAEg(o) → DAEg(c). The dramatic difference in the isomerization quantum yields is ascribed to the formation of an intramolecular charge transfer (ICT)-state for DAEr(o) in polar environments, outcompeting the isomerization reaction[51]. It is acknowledged that such inefficient isomerization process requires extensive light exposure. However, the extremely good photo-stability of the system (see below) implies that the long-term performance is still not affected.

For FRET to occur efficiently, the distance between the donor and the acceptor should be shorter than the critical Förster radius $R_0$, the distance at which the efficiency of the FRET reaction is 50%. Based on a per fluorescence quantum yield of 0.85 in the polymer micelles and assuming $\kappa^2 = 2/3$ (freely rotating chromophores) $R_0$-values of 45 and 51 Å were determined for the per-DAEg(c) and per-DAEr(c) FRET-pairs, respectively (see Table 1

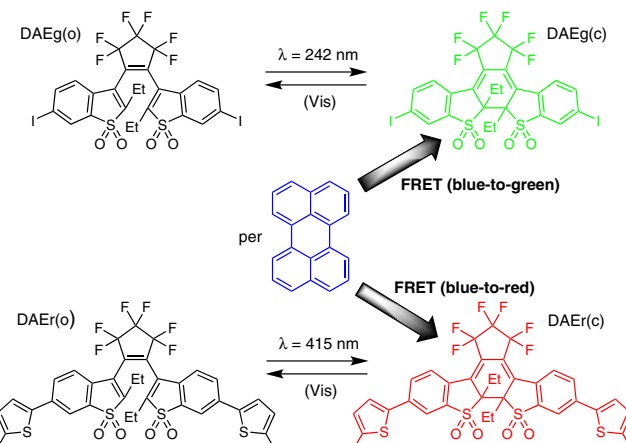

**Fig. 1** Structures and isomerization scheme of the compounds used in this study. The shaded arrows indicate the FRET (Förster Resonance Energy Transfer) processes and the accompanying changes in the emission color. Please note that the photoinduced ring-opening reactions DAE(c) → DAE(o) induced by visible light (Vis) are too inefficient to influence the isomeric distributions with the herein applied irradiation wavelengths

**Table 1 Photophysical properties of the studied compounds. Data determined in the polymer micelles, unless otherwise noted**

| Compound | $\lambda_{max,\ abs}$ (nm)[a] | $\lambda_{max,\ em}$ (nm) | $\Phi_F$ | $\Phi_{iso}$[a, b] | $R_0$ (Å)[c] |
|---|---|---|---|---|---|
| Per | 434 | 443 | 0.85 | – | – |
| DAEg(c) | 435 | 504 | 0.23 | 0.50 | 45 |
| DAEr(c) | 508 | 606 | 0.43 | $5 \times 10^{-4}$ | 51 |

[a]Data in acetonitrile
[b]Isomerization quantum yield for the ring-closing reactions DAE(o) → DAE(c)
[c]Critical Förster distance for the FRET process with per as the donor and DAE(c) as the acceptor

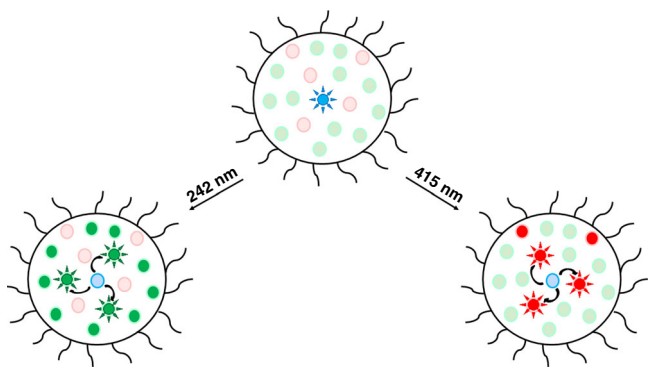

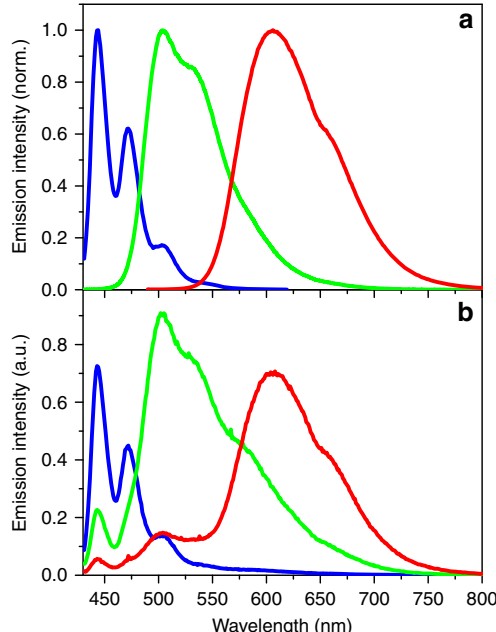

**Fig. 2** Schematic depiction of the micelle cocktail approach used in this study. Curved arrows indicate the orthogonally controlled FRET processes, responsible for the emission color changes

for a compilation of relevant photophysical data). Constraining the three chromophores by covalent means would indeed allow for donor–acceptor (D–A) distances shorter than these $R_0$-values. However, a covalent approach would imply that the three units would have a fixed stoichiometric relation in the structure, and a loss of flexibility in the overall approach. Moreover, the synthetic procedures would be highly complex and tedious. Instead, we opted for a self-assembling cocktail strategy in which the monomeric chromophores are encapsulated in polymer micelles, offering a much larger degree of flexibility as for the composition of the system, which in turn allows a very convenient means of changing both the average D–A distances and the stoichiometry. The overall approach is schematically depicted in Fig. 2.

In addition to the FRET processes indicated in Fig. 2, DAEg(c) and DAEr(c) also constitute a FRET pair with $R_0 = 45$ Å. This observation, however, imposes no obstacles to the intended overall function of the system as long as DAEg can be isomerized with a high degree of selectivity.

**Spectroscopic characterization**. All three chromophores display poor solubility in water, but are dissolved in the presence of the ST-7-8 copolymer[52] (see Supplementary Fig. 1 for the structure, polymer 1b in ref. 52) in aqueous solution by encapsulation into the hydrophobic interior of the polymer micelles, positioning them at distances well suited for efficient FRET reactions to occur[53]. Given the bulk concentrations in our experiments ([per] = 0.45 μM, [DAEg] = 3.7 μM, [DAEr] = 1.1 μM, and [micelles] = 0.31 μM), each micelle contains on the average 1 per, 12 DAEg, and 4 DAEr molecules. This implies average nearest D–A distances compatible with efficient FRET reactions to occur, which is also supported by the experimentally obtained FRET efficiencies of around 0.9 for both the per-DAEg(c) and the per-DAEr(c) bi-component cocktails (see Supplementary Note 4 together with Supplementary Figs. 7 and 9).

Figure 3a shows the normalized emission spectra of per alone, DAEg(c) alone, and DAEr(c) alone in the polymer micelles. The fluorescence quantum yields in this environment were determined to be 0.85 for per, 0.23 for DAEg(c), and 0.43 for DAEr(c). Please note that the quantum yields for the DAE derivatives are dramatically higher compared to all other compounds from the common families of photoswitches, typically displaying fluorescence quantum yields lower than 0.05 in solution. Photographs of the cuvettes under UV irradiation as well as the CIE coordinates of the emission from the respective compound are shown in Fig. 4.

Figure 3b shows the emission spectra from the tri-component cocktail, containing originally per (0.45 μM), DAEg(o) (3.7 μM),

**Fig. 3** Emission spectra in the polymer micelles. **a** Normalized emission spectra of per (blue line), DAEg(c) (green line), and DAEr(c) (red line) in the polymer micelles. **b** Emission spectra of the tri-component micelle cocktail before UV irradiation (blue line), after exposure to 242 nm light (4 nm spectral bandwidth, 10 min irradiation time, green line) and after exposure to 415 nm light (4 nm spectral bandwidth, 2.5 h exposure time, red line). All spectra in panel (**b**) were recorded upon excitation at 423 nm. The relative emission intensities determined by spectral integration in (**b**) are 1.3:4.4:3.7 (blue:green:red), that is, the system displays fairly small intensity variations accompanying the emission color changes

and DAEr(o) (1.1 μM). The blue spectrum was recorded after excitation of the sample at 423 nm. The spectrum is virtually identical to that for per alone, apart from a weak tail at the red end of the spectrum. This tail is ascribed to small amounts of the closed, fluorescent isomers of the DAE units present already at the beginning of the experiment. Comparing the two per emission intensities (Supplementary Fig. 3), it is seen that the intensity dropped by ca. 70% in the cocktail, tentatively ascribed to quenching by electron transfer reactions. The redox data of all three compounds reveal that such reactions are thermodynamically favorable (see Supplementary Note 3). The dynamic nature of the quenching was established by time-resolved single photon counting (SPC) experiments, where the fluorescence lifetime of per decreased from 4.8 to 1.0 ns upon addition of the DAE open isomers.

As anticipated, exposing the sample to light at 242 nm triggers isomerization of mainly DAEg(o) to DAEg(c), and the concomitant FRET reaction from per to DAEg(c). This is apparent by comparing the green spectrum in Fig. 3b ($\lambda_{exc} = 423$ nm) to those of Fig. 3a: The overall emission is dominated by DAEg(c) at the expense of per. Minor emission is also seen from DAEr(c), showing that the 242 nm light is not selective for DAEg(o) to 100%. If the cocktail sample is instead exposed to 415 nm light, the red spectrum is recorded upon excitation at 423 nm. Apart from minor spectral features from per and DAEg(c) at shorter wavelengths, it is clear that the main emitter in this case is DAEr(c). FRET being the dominating quenching mechanism of per is established by comparing the excitation spectra with the corresponding absorption spectra (see Supplementary Fig. 15 and Supplementary Note 5). The CIE

**a**

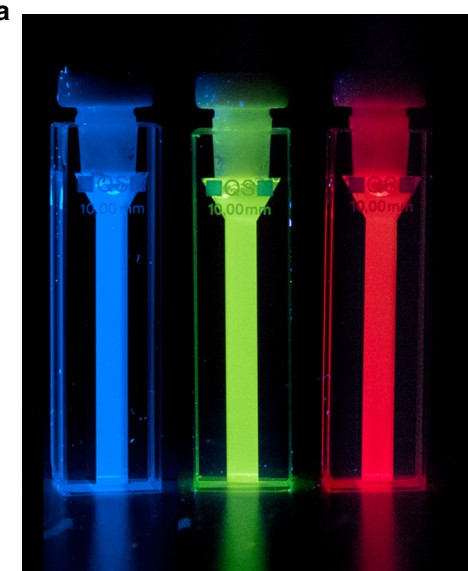

**b**

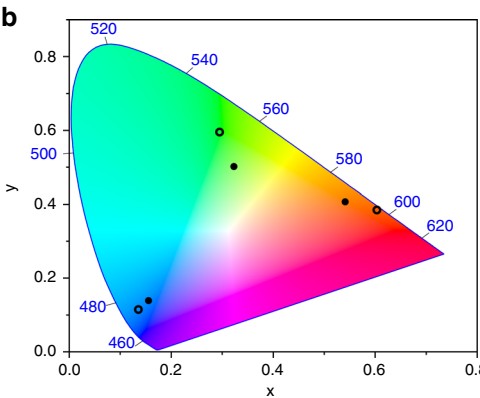

**Fig. 4** Emission colors and CIE coordinates. **a** Photos of the fluorophores alone in the micelles upon excitation at 365 nm. **b** Hollow circles: CIE coordinates for the emission from the fluorophores alone in the micelles: per (0.136, 0.115), DAEg(c) (0.295, 0.596), and DAEr(c) (0.603, 0.385). Solid circles: CIE coordinates for the emission of the tri-component micelle cocktail before UV irradiation (0.156, 0.139), after exposure to 242 nm light (0.323, 0.502) and after exposure to 415 nm light (0.541, 0.407). Please note that all underlying spectra were recorded upon excitation at one and the same wavelength (423 nm). The corresponding coordinates for the sRGB color space are defined as 0.15, 0.06 (blue); 0.30, 0.60 (green); and 0.64, 0.33 (red)

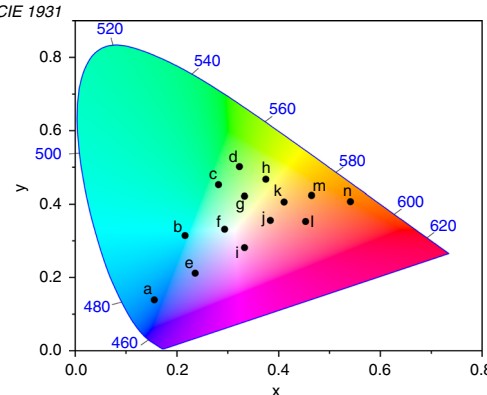

**Fig. 5** Full-color reproduction. CIE coordinates for the emission from the tri-component micelle cocktail after isomerization of both DAE derivatives to different extent. Please note that all underlying spectra were recorded upon excitation at one and the same wavelength (423 nm). Irradiation conditions: **a** no irradiation prior to excitation for emission readout; **b** 220 s. 242 nm; **c** 450 s. 242 nm; **d** 600 s. 242 nm; **e** 15 s. 415 nm; **f** 220 s. 242 nm + 30 s. 415 nm; **g** 450 s. 242 nm + 30 s. 415 nm; **h** 800 s. 242 nm + 30 s. 415 nm; **i** 70 s. 415 nm; **j** 220 s. 242 nm + 120 s. 415 nm; **k** 450 s. 242 nm + 120 s. 415 nm; **l** 370 s. 415 nm; **m** 800 s. 242 nm + 120 s. 415 nm; **n** 2.5 h. 415 nm

component cocktails are collected in Supplementary Figs. 3–9. From here, it is seen that there are slight discrepancies in the quenching efficiencies estimated from steady-state and SPC data, the quenching efficiencies being typically higher from the steady-state experiments. We ascribe this observation to a heterogeneous distribution of the chromophores within the micelles which results in a distribution of the nearest D–A distances. This notion is strengthened by the multi-exponential decays observed for all situations. For the shortest D–A distances, implying a very efficient quenching, the per lifetime is simply too short to be seen in the overall decay which results in an underestimation of the actual quenching efficiency.

Having shown that the tri-component cocktail can generate red, green, and blue emission color, the ultimate test is to see if all colors within the RGB triangle can be generated by isomerization of both DAE derivatives. The cocktail was therefore exposed to a large number of combinations of 242 and 415 nm exposures, and the resulting emission spectra were recorded upon excitation at 423 nm. The corresponding CIE coordinates are shown in Fig. 5. It is very encouraging to note that the performance is as good as anticipated—all colors within the RGB triangle can indeed be reproduced. Adding to the appeal is the fairly small variation in the overall emission intensity for red, green, and blue emission obvious from Fig. 3b, as well as the acceptable absolute emission intensities, where the lowest observed value (blue light) corresponds to a fluorescence quantum yield of around 0.3.

Finally, the system was investigated with respect to photo- and thermal stability (data presented in Supplementary Figs. 10–14). Generally, DAE derivatives are known to be very stable in both these respects, and this has been shown also for the sulfone derivatives studied herein[54–56]. The classical aspect of photo-stability was examined by light-exposure for 160 min or more (365 or 405 nm, 0.7–1.0 mW cm$^{-2}$) of the individual fluorophores encapsulated in the micelles. Per and DAEr showed no signs of decomposition as judged by the absorption and the emission spectra, whereas DAEg experienced an 8% decrease in the absorption and a 25% decrease in the emission intensity upon extensive light exposure. The reduced photostability of the latter compound is tentatively ascribed to an iodo-induced heavy atom

coordinates of the three emission spectra from the cocktail are mapped in the CIE diagram in Fig. 4b. It is encouraging to note how very close the cocktail CIE coordinates are to those of the pure reference samples containing only one of the three fluorophores. We would like to emphasize that all emission spectra are recorded using one and the same excitation wavelength which clearly shows that the dominating reason for the observed color changes is in fact the orthogonally controlled FRET reactions from per to the respective fluorescent DAE isomers. This is further strengthened by the results from SPC experiments. As indicated above, the fluorescence lifetime of per alone in the micelles is 4.8 ns. This number is decreased to 0.90 ns and 0.51 ns in the tri-component cocktails containing mainly per-DAEg(c)-DAEr(o) and per-DAEg(o)-DAEr(c), that is, after exposure to 242 and 415 nm, respectively. Comprehensive steady-state and SPC data including also all possible bi-

effect, increasing the yield of triplet formation which in turn makes it more vulnerable to degrading reactions. Equivalent samples displayed excellent thermal stability as no significant spectral changes were observed in the dark for 20 h (after isomerization of DAEg and DAEr to the closed isomer).

We would also like to emphasize the good color stability of the tri-component cocktail. Direct or FRET sensitized excitations of the DAE photoswitches would indeed trigger isomerization reactions (e.g., DAE(c) → DAE(o)) with concomitant changes in the overall emission spectra[57]. However, due to the very low quantum yields of these processes for the investigated derivatives[51,56], no significant changes in the emission spectra were observed upon exposure to 423 nm light for 65 min—a light dose corresponding to the collection of 100 emission spectra at this excitation wavelength.

## Discussion

We have described a self-assembled all-photonic system consisting of perylene and two molecular photoswitches from the DAE photochromic family where the emission color can be tuned to any color in the RGB system. This is a rare example of stimuli-responsive full-color reproduction, and it is very interesting to note that the system can be operated without the need for physical contact/access. All other previously described examples rely on external stimuli of either physical or mechanical nature, implying that open systems must be used. Accumulation of chemical waste is another problem where chemical stimuli are being applied. Mechanochromic systems undeniably require physical contact which is very unpractical, not to say impossible, for many of the applications foreseen for multicolor fluorescence switching, e.g., bioimaging. Our all-photonic version can be operated remotely, and with unsurpassed spatiotemporal precision, that is, where and when to induce the desired changes in the emission color.

## Methods

**Sample preparation**. All micelle experiments were performed in aqueous solution (mQ). The samples were prepared by mixing amphiphilic copolymer ST-7-8 with the relevant combination of each individual cocktail in dichloromethane and film formation was promoted by evaporation of the solvent in a vacuum chamber. The micelles were formed by rehydration of the film for 10 min. with 2 ml mQ-water followed by filtration with 0.25 μm cellulose acetate filter. The average molecular weight of the individual micelles was determined to be 580 kDa based on static light scattering measurements. With a total amount of 0.4 mg ST-7-8 used in each preparation and a 10% loss in the filtration process, the total micelle concentration was 0.31 μM.

**Spectroscopic measurements**. Ground state absorption spectra were recorded on a Cary 50 UV/vis spectrometer. Corrected fluorescence spectra were recorded on a SPEX Fluorolog-3 spectrofluorometer. Fluorescence lifetimes were measured using a time correlated single photon counting (TC-SPC) setup. The excitation light, $\lambda_{exc} = 405$ nm, was provided at a repetition rate of 20 kHz by a 405 nm diode laser (PicoQuant) and a MCP-PMT detector (10,000 counts in the top channel, 1024 channels). The emitted photons were collected at the magic angle (54.7°) at 440 nm. The measured fluorescence decays were fitted using the program FluoFit Pro v.4 (PicoQuant GmbH, Germany) after deconvolution of the data with the instrument response function (IRF) with FWHM~90 ps.

**Electrochemical characterization**. Solvent and supporting electrolyte: acetonitrile from SDS was used as received (water content <10 mg kg$^{-1}$); tetrabutylammonium tetrafluoroborate or tetrabutylammonium hexafluorophosphate from Sigma-Aldrich were used as a supporting electrolyte without further purification.

A computer controlled VSP potentiostat (Versatile Modular Potentiostat) was used to perform the cyclic voltammetry at low scan rates in the range 0.1–1.0 V s$^{-1}$.

An electrochemical conical cell equipped with a methanol jacket, which makes it possible to set the temperature at 20 °C by means of a thermostat, was used for the set-up of the three electrode system. For cyclic voltammetry experiments, the working electrode was, in all cases, a glassy carbon disk of a diameter of 1.0 mm. It was polished using a 1 mm diamond paste. The counter electrode was a glassy carbon disk of 0.3 cm diameter. All potentials are reported vs. SCE isolated from the working electrode compartment by a salt bridge. The

salt solution of the reference calomel electrode is separated from the electrochemical solution by a salt bridge ended with a frit, which is made of a ceramic material, allowing ionic conduction between the two solutions and avoiding appreciable contamination. Argon was allowed to flow under the solution during the measurements. The concentration of electroactive substances was between 1 and 10 mM while the supporting electrolyte concentration was 0.1 M.

**Isomerization reactions**. The light for selective isomerization of DAEg and DAEr in the polymer micelles at 242 nm and 415 nm was provided by the 450 W xenon lamp in the SPEX Fluorolog-3 spectrofluorometer after passage through the monochromator (4 nm spectral bandwidth). The exposure times were 10 min and 2.5 h, respectively, at photon fluxes of $2.7 \times 10^{-11}$ einstein s$^{-1}$ and $2.7 \times 10^{-9}$ einstein s$^{-1}$ as determined by ferrioxalate actinometry.

**Isomerization quantum yield determination**. The isomerization quantum yields for the reactions DAE(o) → DAE(c) were determined using UV light at 365 nm from an LED (Engin LZ1 10UV00, FWHM = 11 nm, ~1 mW cm$^{-2}$ at the sample). The absorbance changes (DAEg at 450 nm, DAEr at 550 nm) as a function of irradiation time were monitored and compared to those of the reference compounds under identical irradiation power/geometries and corrected for the molar absorption coefficients at 365 nm. The reference compounds used for DAEg and DAEr were diarylethene derivatives **1**[58] and **8**[56].

**Fluorescence quantum yield determination**. For fluorescence quantum yield determination of all compounds in the polymer micelles, relative measurements using a SPEX Fluorolog-3 spectrofluorometer together with standard compounds (perylene in cyclohexane, $\Phi_F = 0.94$; coumarine 153 in deaerated ethanol, $\Phi_F = 0.38$; perylene red in CHCl$_3$, $\Phi_F = 0.96$)[59]. Note that the scattering from the micelles makes it challenging to correctly determine the absorption from the fluorophores at the excitation wavelengths. Herein, we have estimated the contribution from the scattering to the absorption spectra by a linear function between two points in the spectra with zero absorption and zero scattering, respectively. This procedure, yet not exact, should give comparable results to a $\lambda^{-4}$-approach to the scattering.

## Data availability

All relevant data are available from the corresponding author.

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

## Acknowledgements

J.A. acknowledges financial support from the Swedish Research Council VR (Grant # 2016-0360), J.H. and G.G. from MINECO/FEDER through project CTQ2015-65439-R, and F.M.R. from The National Science Foundation (CHE-1505885). Mr. Mats Tiborn is acknowledged for the photography related work. Open access funding provided by Chalmers University of Technology.

## Author Contributions

J.A. designed the study. G.N. performed all the spectroscopic experiments, in some experiments assisted by C.-W. H.J.A. and G.N. interpreted all the spectroscopic data. C.-W.H., M.M., S.T., M.I., H.S., S.-M.L., and F.M.R. contributed to the synthesis of the compounds. J.H. and G.G. performed the electrochemical measurements and the associated data interpretation. J.A. and G.N. wrote the paper.

## Additional information

**Competing interests:** The authors declare no competing interests.

