## [Peer Review File · Nature Communications]

Reviewers' comments:

Reviewer #1 (Remarks to the Author):

Review of "An all-photonic full color RGB system based on molecular photoswitches" by Naren, Andréasson et al...

The manuscript describes a 3 component fluorescent system that when confined in small micelles allows for energy transfer, both FRET and ET, that modify the emission ratios. Through the photochromism of 2 of the components the system can be tuned throughout the RGB spectra.

The concept is of interest, the research is generally well done, though I do believe there are experiments that should be realized to finish the characterization of the system (see below). Additionally I do believe that the manuscript is of interest to the field, yet I do not believe it meets the impact requirement of Nature Communications, I would recommend perhaps Chemical Communications or Advanced Optical Materials.

1) The first question arises from emission stability. The work should show photostability of the system.

a. The classical aspect of photobleaching would change the emission ratios over time. This could be done at the PER wavelength or perhaps the photoswitching wavelengths, etc..

b. The other aspect of emission stability arises from the photochromes and their energy transfer based cycloreversion. Even if the system does not photobleach will the DAE not slowly convert to the open form, shifting your color ratios? This has been shown by the Nakatani and Jovin groups to be of considerable importance. Please see the reference recommendations at the end.

2) The energy transfer from the DAEg and DAEr is never discussed. Though it would appear from the spectra that they would have some spectral overlap and therefore energy transfer.

3) A table with the photophysical parameters of the dyes would be of great use to the reader. These include wavelengths, quantum yields of fluorescence and photoconversion, forster distances, etc..

4) Though not exactly crucial to the manuscript it would be of interest if there was a greater discussion of the micelle formation. Do loading efficiencies change at different concentrations, does it differ by dye, what are the properties of the micelles with only 2 or 1 dye (which you have already done for the PER).

References:

Ouhenia-Ouadahi, K. et al. Fluorescence photoswitching and photoreversible two-way energy transfer in a photochrome-fluorophore dyad. *Photochem. Photobiol. Sci.* 11, 1705–1714 (2012).
Ouhenia-Ouadahi, K. et al. Photochromic-fluorescent-plasmonic nanomaterials: Towards integrated three-component photoactive hybrid nanosystems. *Chem. Commun.* 50, 7299–7302 (2014).
Diaz, S. A., Gillanders, F., Jares-Erijman, E. A. & Jovin, T. M. Photoswitchable semiconductor nanocrystals with self-regulating photochromic Forster resonance energy transfer acceptors. *Nat. Comm.* 6, 6036 (2015).

Reviewer #2 (Remarks to the Author):

The authors demonstrate that color-tuning of fluorescent systems can be achieved over most of the RGB spectral range using an all-photonic approach. The overall emission "color" is constituted of the superposition of the fluorescence of three compounds, two being photochromic.

Developments in fluorescent photochromic molecules were exploited to achieve simultaneous tuneability and fluorescence capabilities, whilst the combination of the necessary three emissive molecules to cover the color-palette is achieved within a simple micellar "cocktail". This allows to control the efficiencies of the necessary energy transfer processes by simple mixing of the compounds. The proper choice of two photochromic emitters is crucial, and is herein appropriate

by exploiting a clear difference in their respective emission colours, and the possibility to orthogonally photoselect them. Indeed, it is possible to photochemically close the red-emitting compound using a specific absorption wavelength of 415 nm, or close the green-emitting compound by exploiting its much larger photoreaction quantum yields combined to a larger absorbance at 242 nm. The simplicity of the system is quite appealing, as well as its all-photonic nature. Conceptually, it constructs onto work previously published by the same authors, for example with cocktails of one donor and one photochromic acceptor achieving blue to white to yellow color tuning (Chem. Sci., 2016, 7, 5867; ref. 28).

Nevertheless, before the manuscript can be considered for publication, the authors should address the various questions that remain open to discussion. A first important limitation of the system appears to be the incapacity to completely cover the RGB range. Indeed, the pure green and red colors are not really reached in the "cocktails". Therefore, is it really correct to call it an RGB system? Is this system actually conceivable as a true RGB system using this type of molecules and cocktails? Can the authors propose a strategy that would enable this?

Another issue is quite inherent to the use of photochromes and the presence of undesired side-reactions. At the photon flux of applications, what is the photostability of the system both in terms of color, and photodegradation due to the electron transfer process? Indeed, it was not discussed in the paper how the irradiation used for induction of fluorescence influences the equilibrium of different forms, although the wavelength used in the visible should clearly open the photochromes over a certain time. This could influence the emission color over time of "use". Additionally, it also has to be noted that the reverse thermal conversion of the photochromes is not discussed at all, although it is of course of known high importance and impact in photochromic systems: once photochemically converted, are the attained chromophore mixtures stable in time in the "cocktails" (over which time-scale)? Prospectively, unless demonstrated experimentally, could the authors propose strategies to "freeze" the colors? In fine, photostability can also be questioned considering the existence of a photo-induced electron transfer process: can this be circumvented by any future strategy? All these issues can be of high importance when mentioning "Bio-imaging" (see conclusions), applications in which the problem of photostability is well-known due to high laser powers and/or long acquisition times.

Other questions are not inherent to the system, but should be clarified:

- Did the authors test a solid-state deposit of the micelles? This might be helpful to evaluate potential future applications.
- What do the authors mean by "concerted isomerization"? Is there an effect of one on the other? Or is it just a simultaneous isomerization?
- Isomerization reactions in the fluorimeter: what were the spectral bandwidths used?
- Quantum yields are averages of 2 methods? Do both estimations differ? If yes, a lot?
- Is steady-state emission quenching of Per calculated at 440 nm emission, or integrated over the whole spectrum?
- Per + DAer(o): how do the authors explain $E(\text{SPC}) > E(\text{SS})$?
- Why not measure the emission quantum yields of the cocktails with the integrating sphere? Since the micelles are absolutely necessary to obtain these systems, it is important to know the impact it would have on the overall emissive efficiency? The authors only extrapolate this from the measurements in acetonitrile. (by the way, in text line 172 it is not clear whether this yield of 0.20 was measured in acetonitrile or other conditions)?
- Line 167: "fairly small variations of intensity": what are the intensity values measured?
- Scheme 1 : both 415 nm and "vis" are in the visible range, the "vis" should be more specific.
- Line 78: "why" should be "while"?
- Line 80: "due to that the isomerization": revise phrase
- Line 82: "virtually selective" = "virtually specific"?, or "highly selective"?
- Line 84: "milieus" ?

Reviewer #3 (Remarks to the Author):

The idea behind this work is brilliant, but its implementation is far from optimized due to the below listed reasons.

- 1- A suitable characterization of the micelles in terms of size and overall content is not provided. At the local concentration estimated inside the polymer micelles (around $1E-3$), per should exhibit excimer emission. No such emission is detected and no comments are made to explain why.
- 2- No mention is made of irradiation times. This is crucial in light of the vastly different isomerization quantum yields of the photoreactions, which must introduce a strong degree of temporal "asymmetry" of the two processes under similar irradiation conditions.
- 3- According to the experimental setup described (spectrofluorimeter also used as photoreactor), it seems that simultaneous irradiation of the two DAEs is not made, which substantially limits the potential of the three-component cocktail that has been designed.
- 4- In Scheme 1 the back reactions are too generically indicated as far as light excitation is concerned. This is a crucial piece of information to account for the reliability/stability of the system under the reading mode at 423 nm.
- 5- Although the system is presented as clean and efficient, the supposed occurrence of electron transfer quenching between per and the two switches introduces an undue level of interference and complexity.
- 6- The selection of 423 nm as reading wavelength is not rationally explained.
- 7- The presence of multiexponential excited state decays is rationalized with heterogeneous distributions inside the micelles. Authors should instead more carefully examine the complexity of the systems, as pointed out in some points above, which may imply the presence of multiple quenching processes.
- 8- Figure 3 is scarcely meaningful, with dots reported in a CIE diagram without indicating the exact experimental conditions in which each of them were obtained.

This manuscript cannot be considered for publication in Nature Communications.

Reviewer #1

Q1. The first question arises from emission stability. The work should show photostability of the system.

a. The classical aspect of photobleaching would change the emission ratios over time. This could be done at the PER wavelength or perhaps the photoswitching wavelengths, etc.

A1a. Generally, DTE derivatives are known for their extreme photostability, and DAEr reported herein has been explicitly studied before in this context (see reference below). Here, the classical aspect of photobleaching for the three individual compounds (perylene, DAEG, and DAEr) in acetonitrile and encapsulated in the micelles were investigated in the following experiments:

For perylene, the samples were exposed to 405 nm LED light (1mW/cm²) for 160 min. The absorption and the emission of the sample was recorded prior to and after the irradiation. No significant changes between these spectra were observed, apart from small changes in the scattering component to the absorption spectra in the micelles. Data shown in the Supplementary Information ("Photostability").

For DAEr, the sample was first isomerized to 100% closed isomer by exposure to 365 nm UV light (0.7 mW/cm²). Extended irradiation with this light-source for 160 min did not result in any significant changes in the absorption or the emission intensity, apart from small changes in the scattering component to the absorption spectra in the micelles. This excellent photostability has been shown before (J. Am. Chem. Soc. 2011, 133, 13558-13564). Data shown in the Supplementary Information ("Photostability").

For DAEG, the sample was first isomerized to 100% closed isomer by exposure to 365 nm UV light (0.7 mW/cm²). Extended irradiation with this light-source for 285 min (in acetonitrile) and 160 min (in the polymer micelles) resulted in a significant (between 6% and 25%) decrease in the absorption and in the emission intensity. The typical DAE sulfone derivative is much more photostable than this. We ascribe the reduced photostability of this particular derivative to the iodo-induced heavy atom effect, increasing the yield of triplet formation which in turn makes the molecule more vulnerable to degradation. Data shown in the Supplementary Information. ("Photostability").

In addition to the figures added to the Supplementary Information, we have also added a final section to the manuscript that describes all the above results on the long term stability.

Q1b. The other aspect of emission stability arises from the photochromes and their energy transfer based cycloreversion. Even if the system does not photobleach will the DAE not slowly convert to the open form, shifting your color ratios? This has been shown by the Nakatani and Jovin groups to be of considerable importance. Please see the reference recommendations at the end.

A1b. Indeed, FRET-induced sensitization of the opening reactions for the DAE derivatives would have been an issue if it was not for the very low quantum yields of these reactions. For DAEr, we can be no more specific than that the yield is lower than 10⁻⁵, (DyePig 2018, 153, 144-149) and no attempts were made to put more accurate numbers on DAEG. In order to demonstrate the resistance toward light- and FRET-induced isomerizations, a tri-component cocktail was prepared and isomerized to contain both the open and the closed isomer of the two DAE derivatives. Upon exposure at 423 nm (excitation wavelength for emission readout) for 65 min (corresponding to recording 100 spectra with the actual instrument setting) did not change the absorption or the emission spectra significantly. It follows that the CIE coordinates and the emission color for all recorded spectra are the same (data shown in the

Supplementary Information, "Color stability"). We have also included these arguments in the manuscript text (last section), and a reference is given to Nat. Comm. 6, 6036 (2015) as suggested.

Q2. The energy transfer from the DAEg and DAEr is never discussed. Though it would appear from the spectra that they would have some spectral overlap and therefore energy transfer.

A2. This is a very valid observation. The R_0 -value for this D-A pair is 45 Å, and it is also experimentally observed that DAEr(c) is efficiently quenching the emission from DAEg(c) in the micelles ($E_{FRET} = 82\%$ for the bi-component cocktail DAEg(c)-DAEr(c)). This fact illustrates even more the importance of high selectivity in the open-to-closed isomerization reactions. We have added this information to the manuscript text.

Q3. A table with the photophysical parameters of the dyes would be of great use to the reader. These include wavelengths, quantum yields of fluorescence and photoconversion, forster distances, etc.

A3. The requested table has been added to the manuscript.

Q4. Though not exactly crucial to the manuscript it would be of interest if there was a greater discussion of the micelle formation. Do loading efficiencies change at different concentrations, does it differ by dye, what are the properties of the micelles with only 2 or 1 dye (which you have already done for the PER).

A4. Earlier investigations in our laboratories (Langmuir 2016, 32, 8676–8687, cited in the manuscript) demonstrated that the loading efficiency for a given dye can be regulated simply by adjusting the relative amounts of amphiphilic polymer and guest chromophore. The solubility of the dye in water controls predominantly the number of guests encapsulated in each micellar host. As a result, we expect the loading efficiency to change with the structure of the dye, although we have not yet investigated systematically these effects.

Reviewer #2

Q1. A first important limitation of the system appears to be the incapacity to completely cover the RGB range. Indeed, the pure green and red colors are not really reached in the "cocktails". Therefore, is it really correct to call it an RGB system? Is this system actually conceivable as a true RGB system using this type of molecules and cocktails? Can the authors propose a strategy that would enable this?

A1. There are several color systems that define the CIE coordinates of R, G, and B, and there are substantial discrepancies between these systems. This is why it is hard to say that we do or we don't generate the "pure colors" or the colors of a "true RGB system". The reviewer's question if we can propose a strategy that would enable a true RGB system is a bit tricky to answer, as there is no single "true RGB system".

Q2. Another issue is quite inherent to the use of photochromes and the presence of undesired side-reactions. At the photon flux of applications, what is the photostability of the system both in terms of color, and photodegradation due to the electron transfer process? Indeed, it was not discussed in the

paper how the irradiation used for induction of fluorescence influences the equilibrium of different forms, although the wavelength used in the visible should clearly open the photochromes over a certain time. This could influence the emission color over time of “use”.

A2. See A1a and A1b for Reviewer 1 above.

Q3. Additionally, it also has to be noted that the reverse thermal conversion of the photochromes is not discussed at all, although it is of course of known high importance and impact in photochromic systems: once photochemically converted, are the attained chromophore mixtures stable in time in the “cocktails” (over which time-scale)? Prospectively, unless demonstrated experimentally, could the authors propose strategies to “freeze” the colors?

A3. *The typical DAE derivative displays very slow thermal reaction from the closed to the open isomer, and this is true also for DAEg(c) and DAEr(c) used here. The absorption of both these forms were monitored, both in acetonitrile and in the micelles, for 20 h without any detectable changes. This information has been added to the manuscript text, and supporting figures to the Supplementary Information (“Thermal stability”).*

Q4. In fine, photostability can also be questioned considering the existence of a photo-induced electron transfer process: can this be circumvented by any future strategy? All these issues can be of high importance when mentioning “Bio-imaging” (see conclusions), applications in which the problem of photostability is well-known due to high laser powers and/or long acquisition times.

A4. *We did not expose our cocktail system to any fluorescence microscopy measurements, but there are previously reported systems based on molecular dyads containing similar sulfone DAE:s that display photo-induced electron transfer (PET) reactions. These systems were proven to be stable enough to allow for extended single molecule detection (J. Am. Chem. Soc. 2011, 133, 13, 4984-4990). We have added this reference to the manuscript.*

Other questions are not inherent to the system, but should be clarified:

Q5. Did the authors test a solid-state deposit of the micelles? This might be helpful to evaluate potential future applications.

A5. *In principle, it should be possible to record emission spectra of drop-cast deposits of solutions containing the loaded micelles. We expect, however, that the discrete supramolecular assemblies will fuse into large aggregates with the slow evaporation of the solvent. It is hard to predict how the encapsulated dyes will distribute in the resulting material. We will design appropriate model experiments to unravel these effects with systematic investigations and report these results in a future publication.*

Q6. What do the authors mean by “concerted isomerization”? Is there an effect of one on the other? Or is it just a simultaneous isomerization?

A6. *We have taken out “concerted” as it indeed may lead the reader to believe that isomerization of one photoswitch will have an effect on the other.*

Q7. Isomerization reactions in the fluorimeter: what were the spectral bandwidths used?

A7. *A spectral bandwidth of 4.0 nm was used. This information has been added to the legend of Figure 1 and the Supplementary Information (“Isomerization reactions”).*

Q8. Quantum yields are averages of 2 methods? Do both estimations differ? If yes, a lot?

A8. In the revised version of the manuscript, we report the fluorescence quantum yields inside the polymer micelles measured only by the comparative method.

Q9. Is steady-state emission quenching of Per calculated at 440 nm emission, or integrated over the whole spectrum?

A9. The quenching efficiency of Per is calculated by comparing the emission intensities at 443 nm. Integration over the whole spectrum would not be meaningful, as there is substantial spectral overlap with the emission from DAEg(c).

Q10. Per + DAer(o): how do the authors explain $E(\text{SPC}) > E(\text{SS})$?

A10. We write that “the quenching efficiencies being typically higher from the steady-state experiments”, so we are not claiming they always are. The reason why this notion does not hold in this particular case is unclear to the authors.

Q11. Why not measure the emission quantum yields of the cocktails with the integrating sphere? Since the micelles are absolutely necessary to obtain these systems, it is important to know the impact it would have on the overall emissive efficiency? The authors only extrapolate this from the measurements in acetonitrile. (by the way, in text line 172 it is not clear whether this yield of 0.20 was measured in acetonitrile or other conditions)?

A11. See A8 above. The reason for using the comparative method (using a standard) is that the integrating sphere has been contaminated. The value referred to above as “0.20” has now been changed to correspond to the actual quantum yield in the polymer micelles.

Q12. Line 167: “fairly small variations of intensity”: what are the intensity values measured?

A12. The relative intensities of the Blue:Green:Red emission are 1.3:4.4:3.7 based on spectral integration. We have included the integrated values to the legend of Figure 1.

Q13. Scheme 1 : both 415 nm and “vis” are in the visible range, the “vis” should be more specific.

A13. See A4 for Reviewer 3 below.

Q14. Line 78: “why” should be “while”?

A14. “why” has been changed to “so that”.

Q15. Line 80: “due to that the isomerization”: revise phrase

A15. “due to that” has been changed to “as”.

Q16. Line 82: “virtually selective” = “virtually specific”?, or “highly selective”?

A16. Oops... “virtually selective” has been changed to “virtually specific”.

Q17. Line 84: “milieus”?

A17. “milieus” has been changed to “environments”.

Reviewer #3

Q1. A suitable characterization of the micelles in terms of size and overall content is not provided. At

the local concentration estimated inside the polymer micelles (around $1E-3$), per should exhibit excimer emission. No such emission is detected and no comments are made to explain why.

A1. We agree with the reviewer that this data can be presented in a clearer fashion. Although the data described above was included in the original version, it was scattered over the manuscript text and the Supplementary Information. We have now collected all this data in the section "Estimation of the donor-acceptor distances in the micelles" in the Supplementary Information. To specifically answer the question about the absence of excimer emission of perylene from inside the micelles, the majority of all individual micelles are expected to contain only one perylene molecule (which has been added to the manuscript text). Thus, we have put less emphasis on the actual intra-micellar concentrations in the revised version of the manuscript.

Q2. No mention is made of irradiation times. This is crucial in light of the vastly different isomerization quantum yields of the photoreactions, which must introduce a strong degree of temporal "asymmetry" of the two processes under similar irradiation conditions.

A2. The irradiation times are listed in the Supplementary Information under "Isomerization reactions", but we have now also included this data in the legend of Figure 1 in the manuscript.

Q3. According to the experimental setup described (spectrofluorimeter also used as photoreactor), it seems that simultaneous irradiation of the two DAEs is not made, which substantially limits the potential of the three-component cocktail that has been designed.

A3. Indeed, in the present study, we could not isomerize both DAE derivatives at the same time as we are using one and the same Xe-lamp for these purposes. We believe that this is a technical point that could "easily" be resolved with an additional Xe-lamp and a monochromator. We would also like to stress that the present system is a proof-of-principle rather than a presentation of a ready-to-use device.

Q4. In Scheme 1 the back reactions are too generically indicated as far as light excitation is concerned. This is a crucial piece of information to account for the reliability/stability of the system under the reading mode at 423 nm.

A4. We agree that Scheme 1 could be improved. "Vis" has been put in parenthesis to indicate that the light-induced decolorization reaction is very inefficient. We have also added to the legend that "Please note that the isomerization reactions closed-to-open (c)→(o) induced by visible light (Vis) are too inefficient to influence the isomeric distributions with the herein applied irradiation wavelengths".

Q5. Although the system is presented as clean and efficient, the supposed occurrence of electron transfer quenching between per and the two switches introduces an undue level of interference and complexity.

A5. The optimal system should, of course, not display any PET reactions at all. However, as soon as the DAE derivatives are isomerized to the respective closed form, the efficiencies of the FRET reactions are high enough to outcompete the PET reactions (as judged by the excitation spectra and the corresponding absorption spectra, data not shown). Thus, the only negative impact from PET is that the fluorescence intensity from perylene is quenched when both DAE photoswitches are in the open isomeric form (before any light exposure).

Q6. The selection of 423 nm as reading wavelength is not rationally explained.

A6. 423 nm was selected as to differentiate (red shifting) the excitation wavelength for emission readout as much as possible from the 242 nm and 415 nm light, at the same time avoiding influence from scattering in the emission spectra. This information has been added to the Supplementary Information (Steady-state emission and time resolved single photon counting (SPC) data of the micelle cocktails).

Q7. The presence of multiexponential excited state decays is rationalized with heterogeneous distributions inside the micelles. Authors should instead more carefully examine the complexity of the systems, as pointed out in some points above, which may imply the presence of multiple quenching processes.

A7. As indicated above (A5), as far as we are concerned, the only downside with multiple quenching processes (PET in addition to FRET) is the undesired initial quenching of perylene. The presence of multiple quenching reactions per se will not cause multiexponential decays in the absence of heterogenous distributions.

Q8. Figure 3 is scarcely meaningful, with dots reported in a CIE diagram without indicating the exact experimental conditions in which each of them were obtained.

A8. Indeed, to make Figure 3 more meaningful we have added the irradiation conditions for each of the CIE coordinates to the Supplementary Information (Figure S15).

Finally, we have added the following paper to the list of references: *Angew. Chem. Int. Ed.* 2019, 58, 3082–3086. Moreover, we realized that we used a lightly different polymer from what was described in the original submission (corrected for in the re-submission, ST-7-4 substituted for ST-7-8). However, this has no implications whatsoever for the experimental results or the interpretation thereof.

Reviewers' comments:

Reviewer #1 (Remarks to the Author):

The authors have addressed all the issues I brought up and I believe the manuscript should be published as is. I note that my impact comment was not answered by the authors, I will drop the issue and leave it to the editors guidance.

Reviewer #2 (Remarks to the Author):

The authors have improved the manuscript by demonstrating *de facto* that the system studied is reasonably photostable and thermally stable. The added precisions were necessary to render the manuscript publishable. The manuscript does not present any major flaw anymore and clearly supports the RGB color tuning strategy.

Although this remains acceptable, it has to be however noticed that :

- Although the experimental conditions of irradiation have been given with more precision, the irradiation in a fluorimeter cannot be reproduced exactly in another laboratory (although power and bandwidth are given, it depends on quality of optics, alignment, ...). Actinometry (or another power measurement experiment), used for the new stability experiments, would also have improved the reproducibility of the color-tuning experiments.
- The authors have mentioned the existence of several RGB systems, why not cite the most closely approached by this data? (My point was actually more related to the fact that the R seems more "orangy" than red...).
- Some aspects are not discussed (unless I've overseen it), such as the impact of weak isomerization yield of DAer and thus possible long irradiation times for the "switching", or the emission spectral width (only the "color" is discussed. The authors mention bioimaging, in that case a large spectral width would generally be an undesired property).

Reviewer #3 (Remarks to the Author):

The authors have made some efforts to improve the paper. However, the following points make it still unsuitable for the top standards of Nat Commun.

- 1- A quantitative calculation/distribution of the concentration of Per molecules inside the micelles must be reported. Stating "the majority of all individual micelles are expected to contain only one perylene molecule" is not scientifically acceptable. Now the authors claim that there is typically one Per molecule inside each micelle. This is not in line with the cartoon in Scheme 1, where several Per molecules are reported inside the micelles, which creates confusion.
- 2- Apparently, no quantitative data are reported on the kinetics of PET and FRET. In other words, there is no clear experimental evidence supporting the conclusion "the efficiencies of the FRET reactions are high enough to outcompete the PET reactions".
- 3- What is probably the most important experimental information is segregated in the last section of the Supp Info as "Annotated multicolor CIE diagram", whereas a poorly significant CIE dotted diagram is reported in the main text as Fig. 3. Why?

Reviewer #2

The authors have improved the manuscript by demonstrating *de facto* that the system studied is reasonably photostable and thermally stable. The added precisions were necessary to render the manuscript publishable. The manuscript does not present any major flaw anymore and clearly supports the RGB color tuning strategy.

Although this remains acceptable, it has to be however noticed that:

Q1. Although the experimental conditions of irradiation have been given with more precision, the irradiation in a fluorimeter cannot be reproduced exactly in another laboratory (although power and bandwidth are given, it depends on quality of optics, alignment, ...). Actinometry (or another power measurement experiment), used for the new stability experiments, would also have improved the reproducibility of the color-tuning experiments.

A1. We for sure agree that it would have been very helpful if it was as easy as just giving the photon fluxes at 242 nm and 415 nm. However, just like the reviewer is pointing out, differences in lamp intensities, optics, geometries etc. make every fluorimeter unique. Thus, if we would have reported our photon flux at the given slit width (spectral resolution), another fluorimeter is very likely to display a different spectral resolution at the given photon flux (slit width and spectral resolution cannot be changed independently). This is why we chose to not report on the photon flux. Instead we believe that providing the isomerization quantum yields for the closing reactions of the two DAE derivatives is more useful information.

Q2. The authors have mentioned the existence of several RGB systems, why not cite the most closely approached by this data? (My point was actually more related to the fact that the R seems more “orangy” than red...).

A2. We have added the RGB coordinates for the most commonly adopted sRGB system to the legend of Figure 2.

Q3. Some aspects are not discussed (unless I’ve overseen it), such as the impact of weak isomerization yield of DAER and thus possible long irradiation times for the “switching”, or the emission spectral width (only the “color” is discussed. The authors mention bioimaging, in that case a large spectral width would generally be an undesired property).

A3. The above mentioned drawbacks with low isomerization quantum yield for DAER have been added to the manuscript.

Reviewer #3

The authors have made some efforts to improve the paper. However, the following points make it still unsuitable for the top standards of Nat Commun.

Q1. A quantitative calculation/distribution of the concentration of Per molecules inside the micelles

must be reported. Stating “the majority of all individual micelles are expected to contain only one perylene molecule” is not scientifically acceptable. Now the authors claim that there is typically one Per molecule inside each micelle. This is not in line with the cartoon in Scheme 1, where several Per molecules are reported inside the micelles, which creates confusion.

A1. It is written in the manuscript text that “Given the bulk concentrations in our experiments ($[per] = 0.45 \mu\text{M}$, $[DAEg] = 3.7 \mu\text{M}$, $[DAEr] = 1.1 \mu\text{M}$, and $[micelles] = 0.31 \mu\text{M}$), each micelle contains on the average 1 per, 12 DAEg, and 4 DAEr molecules.” We believe that this is a quantitative calculation that clearly supports our statement that there is only one perylene inside each micelle on the average. The distribution of molecules inside the micelles is by no means static, as individual micelles exchange materials with one another in a dynamic fashion.

We are sorry for having misled the referee to interpret our schematic representation of the micelles in Scheme 1 too quantitatively. We have changed the cartoon to give a better representation of the true situation.

Finally, we have added some text to the Supplementary Information (Estimation of the donor-acceptor distances in the micelles) that the absence of excimer emission from perylene inside the micelles supports that there is only one perylene in each micelle. Herein, we have also given the concentrations that this situation would correspond to.

Q2. Apparently, no quantitative data are reported on the kinetics of PET and FRET. In other words, there is no clear experimental evidence supporting the conclusion “the efficiencies of the FRET reactions are high enough to outcompete the PET reactions”.

A2. A comparison between the excitation and the absorption spectra (the emission monitored in the spectral region where only the DAEC derivatives emit) of the bi-component cocktails per-DAEg(c) and per-DAEr(c) clearly shows that the dominating quenching mechanism is FRET, rather than PET. We have included this data in the resubmission. Text in the manuscript, and Figures in the Supplementary Information (Figure S15).

Q3. What is probably the most important experimental information is segregated in the last section of the Supp Info as “Annotated multicolor CIE diagram”, whereas a poorly significant CIE dotted diagram is reported in the main text as Fig. 3. Why?

A3. We believe that the CIE diagram is much clearer and less busy in the absence of the annotations. The interested reader could look up the details in the Supplementary Information. Anyway, we have decided to follow the referee’s suggestion to include the annotated figure in the manuscript.

Reviewers' comments:

Reviewer #2 (Remarks to the Author):

The authors have responded to the issues reported. However, and I'm sorry to go back and forth on this, the following response triggers an additional comment:

"(...) Thus, if we would have reported our photon flux at the given slit width (spectral resolution), another fluorimeter is very likely to display a different spectral resolution at the given photon flux (slit width and spectral resolution cannot be changed independently). This is why we chose to not report on the photon flux."

This suggests that the authors know the photon-flux. So, me and probably other readers would like to be informed about the photon-flux, if it is known. The knowledge of the combination of photon-flux and spectral width allows to reproduce the irradiation conditions by other research groups, with setups that are not necessarily a fluorimeter. In contrast, as the authors themselves mention, the fluorimeter irradiation conditions are not exactly reproducible. The fluorimeter irradiation conditions could be acceptable at this stage of the reviewing, if the photon-flux was NOT known (but that again is in contradiction with the authors reply).

Reviewer #3 (Remarks to the Author):

N/A

Reviewer #2

Q1. The authors have responded to the issues reported. However, and I'm sorry to go back and forth on this, the following response triggers an additional comment:

"(...) Thus, if we would have reported our photon flux at the given slit width (spectral resolution), another fluorimeter is very likely to display a different spectral resolution at the given photon flux (slit width and spectral resolution cannot be changed independently). This is why we chose to not report on the photon flux."

This suggests that the authors know the photon-flux. So, me and probably other readers would like to be informed about the photon-flux, if it is known. The knowledge of the combination of photon-flux and spectral width allows to reproduce the irradiation conditions by other research groups, with setups that are not necessarily a fluorimeter. In contrast, as the authors themselves mention, the fluorimeter irradiation conditions are not exactly reproducible. The fluorimeter irradiation conditions could be acceptable at this stage of the reviewing, if the photon-flux was NOT known (but that again is in contradiction with the authors reply).

A1. No worries, there are indeed very likely many readers that will find the information on the photon fluxes valuable. We have included these numbers in the Supplementary Information under "Isomerization reactions" in the Experimental section. We have also included suitable references to the method by which the fluxes were determined (ferrioxalate actinometry).

REVIEWERS' COMMENTS:

Reviewer #2 (Remarks to the Author):

The authors have responded to the comment and provided the photon fluxes.
Two additional references were reported in the SI, although the format of ref.4 may need to be checked.

REVIEWERS' COMMENTS:

Reviewer #2 (Remarks to the Author):

The authors have responded to the comment and provided the photon fluxes.
Two additional references were reported in the SI, although the format of ref.4 may need to be checked.

Answer: This has been taken care of.